# Application of Various Metal-Organic Frameworks (MOFs) as Catalysts for Air and Water Pollution Environmental Remediation

**Sanha Jang [1], Sehwan Song [2], Ji Hwan Lim [3], Han Seong Kim [3], Bach Thang Phan [4], Ki-Tae Ha [5], Sungkyun Park [2,*] and Kang Hyun Park [1,*]**

[1] Department of Chemistry, Pusan National University, Busandaehak-ro 63beon-gil, Geumjeong-gu, Busan 46241, Korea; jangs0522@naver.com

[2] Department of Physics, Pusan National University, Busandaehak-ro 63beon-gil, Geumjeong-gu, Busan 46241, Korea; sehwan465@gmail.com

[3] Department of Organic Material Science and Engineering, Pusan National University, Geumjeong-gu, Busan 46241, Korea; wdwf455@naver.com (J.H.L.); hanseongkim@pusan.ac.kr (H.S.K.)

[4] Center for Innovative Materials and Architectures (INOMAR), Vietnam National University, Ho Chi Minh City 721337, Vietnam; pbthang@inomar.edu.vn

[5] Department of Korean Medical Science, Healthy Aging Korea Medical Research Center, Pusan National University, Yangsan 50612, Gyeongsangnam-do, Korea; hagis@pusan.ac.kr

**\*** Correspondence: psk@pusan.ac.kr (S.P.); chemistry@pusan.ac.kr (K.H.P.); Tel.: +82-51-510-2238 (K.H.P.)

**Abstract:** The use of metal-organic frameworks (MOFs) to solve problems, like environmental pollution, disease, and toxicity, has received more attention and led to the rapid development of nanotechnology. In this review, we discuss the basis of the metal-organic framework as well as its application by suggesting an alternative of the present problem as catalysts. In the case of filtration, we have developed a method for preparing the membrane by electrospinning while using an eco-friendly polymer. The MOFs were usable in the environmental part of catalytic activity and may provide a great material as a catalyst to other areas in the near future.

**Keywords:** metal organic framework; environmental pollution; filter; gas sorption; sensor; hydrogen storage; electrospinning

## 1. Introduction

Recently, environmental pollution is increasing due to toxic waste and hazardous organic compounds [1]. The amount of poisonous compounds that are released into the environment is a serious problem for human life. In the pragmatic situation, air and organic pollutants are highly involved they can be commonly expressed as particulates, acidic substances, gases, or mixtures [2,3].

Nitroaromatic compounds are found in soil, air, and water samples due to wastewater sources from the plastic, pesticide, pharmaceutical, and dye industry [4]. In addition, the development of hydrogen and methane storage systems is necessary for the widespread use of green energy. Moreover, the separation and selective gas adsorption of poisonous gases, such as nitrogen dioxide and ammonia gas, are significant in the field of air pollution [5–9].

It is essential that sensors should be developed as the way to prevent toxic materials, one of the leading causes of environment pollution, from being released into the environment. Sulfur dioxide and nitroaromatic compounds are considered to be poisonous and harmful to human life [10]. Our previous work has reported a PdAg nanoparticle infused metal-organic framework (MOF) used for

the detection of 4-nitrophenol while using an electrochemical sensor [11]. In addition, Salama et al., reported an $SO_2$ gas sensor while using an MOF [12].

Metal-organic frameworks (MOFs) have a wide range of interesting properties such as high specific surface area and facile modification [13–26]. MOFs are microporous materials that form three-dimensional (3D) crystalline networks, which are prepared by combining various metal ions with organic linkers in an appropriate solvent [27–32]. Over the past years, MOFs have been used as catalysts, absorbents, and filters. MOFs have many advantages due to their modifiable properties, such as high specific surface area and porosity [30,33–36]. The exceptional characteristics of MOFs have led to their possible application in a wide range of technological areas, including gas sorption, separation, storage [5–9], sensing [10–12,37], and heterogeneous catalysis [38–53].

In this review, we focus on various MOFs and their applications in different fields, such as: (1) controlled gas uptake of toxic gases, including ammonia, nitrogen dioxide, and sulfur dioxide [6,7,12]; (2) sensors using PdAg nanoparticle infused MOFs and Cd-MOFs [11,54]; (3) Hydrogen gas storage [55–57]; and, (4) filtration for air and water pollution control while using nanofibrous MOFs [58,59] (Scheme 1).

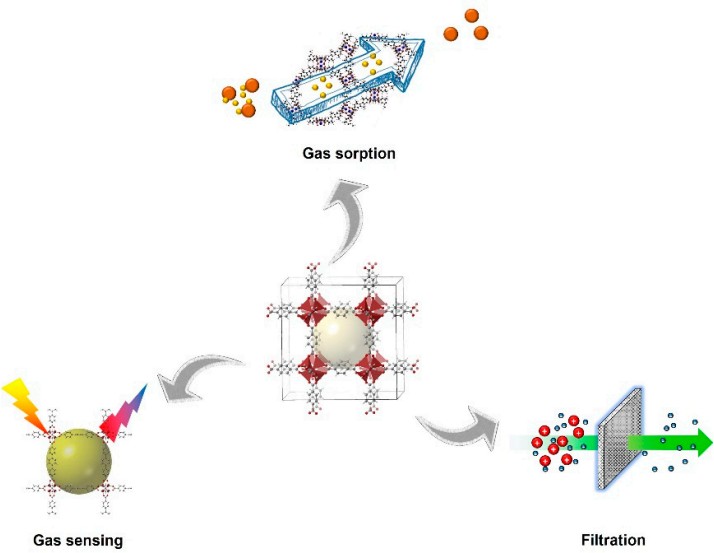

**Scheme 1.** Various applications by metal-organic frameworks (MOFs).

## 2. Basic of Stable MOFs

### 2.1. Characterization of MOFs

MOFs are an arising group of porous materials that were synthesized from metal ions and organic linkers [60,61]. The ever-expanding improvement in the performance of MOFs and facile control over their properties, MOFs have attracted the interest of engineers and scientists [62–67]. Earlier MOFs were synthesized from divalent metals, which showed superior properties and a diverse range of applications [60], such as gas sorption, separation, storage [5–9], sensing [10–12,37], and heterogeneous catalysis [38–53]. In particular, stable MOFs can be predicted by the strength of the metal-organic linker bond that formed in their framework. We have summarized some of the typical stable MOFs prepared from divalent, trivalent, and tetravalent metal ions in Table 1 [68].

**Table 1.** Abridgment of some stable MOFs [68].

| MOFs [a] | Clusters/Cores | Linkers [b] | BET Surface Area ($m^2 \, g^{-1}$) | Ref. |
|---|---|---|---|---|
| MIL-53 (Al) | $[Al(OH)(COO)_2]_n$ | BDC | 1181 | [69] |
| Al-FUM | $[Al(OH)(COO)_2]_n$ | FUM | 1080 | [70,71] |
| MIL-69 | $[Al(OH)(COO)_2]_n$ | 2,6-NDC | NA | [72] |

| | | | | |
|---|---|---|---|---|
| MIL-96 (Al) | $[Al_3(\mu_3\text{-}O)(COO)_6]$ $[Al(OH)(COO)_2]_n$ | BTC | NA | [73] |
| MIL-100 (Al) | $[Al_3(\mu_3\text{-}O)(COO)_6]$ | BTC | 2152 | [74] |
| MIL-101 (Al) | $[Al_3(\mu_3\text{-}O)(COO)_6]$ | BDC-NH$_2$ | 2100 | [75] |
| MIL-110 | $[Al_8(OH)_{15}(COO)_9]$ | BTC | 1400 | [76] |
| MIL-118 | $[Al(OH)(COO)_2(COOH)_2]_n$ | BTEC | NA | [77] |
| MIL-120 | $[Al(OH)(COO)_2]_n$ | BTEC | 308 | [78] |
| MIL-121 | $[Al(OH)(COO)_2]_n$ | BTEC | 162 | [79] |
| MIL-122 | $[Al(OH)(COO)_2]_n$ | NTC | NA | [80] |
| DUT-5 | $[Al(OH)(COO)_2]_n$ | BPDC | 1613 | [81] |
| NOTT-300 | $[Al(OH)(COO)_2]_n$ | BPTA | 1370 | [82] |
| CAU-1 | $[Al_8(OH)_4(OCH_3)_8(COO)_{12}]$ | BDC-NH$_2$ | 1700 [c] | [83] |
| CAU-3-BDC | $[Al_{12}(OCH_3)_{24}(COO)_{12}]$ | BDC | 1550 | [84] |
| CAU-3-BDC-NH$_2$ | $[Al_{12}(OCH_3)_{24}(COO)_{12}]$ | BDC-NH$_2$ | 1250 | [84] |
| CAU-3-NDC | $[Al_{12}(OCH_3)_{24}(COO)_{12}]$ | 2,6-NDC | 2320 | [84] |
| CAU-4 | $[Al(OH)(COO)_2]_n$ | BTB | 1520 | [85] |
| CAU-8 | $[Al(OH)(COO)_2]_n$ | BeDC | 600 | [86] |
| CAU-10 | $[Al(OH)(COO)_2]_n$ | 1,3-BDC | 635 | [87] |
| 467-MOF | $[Al(OH)(COO)_2]_n$ | BTTB | 725 | [88] |
| Al-PMOF | $[Al(OH)(COO)_2]_n$ | TCPP | 1400 | [89] |
| PCN-333 (Al) | $[Al_3(\mu_3\text{-}O)(COO)_6]$ | TATB | 4000 | [90] |
| PCN-888 (Al) | $[Al_3(\mu_3\text{-}O)(COO)_6]$ | HTB | 3700 | [91] |
| Al-soc-MOF-1 | $[Al_3(\mu_3\text{-}O)(COO)_6]$ | TCPT | 5585 | [92] |
| MIL-53 (Cr) | $[Cr(OH)(COO)_2]_n$ | BDC | NA | [93] |
| MIL-88A (Cr) | $[Cr_3(\mu_3\text{-}O)(COO)_6]$ | FUM | NA | [94] |
| MIL-88B (Cr) | $[Cr_3(\mu_3\text{-}O)(COO)_6]$ | BDC | NA | [94] |
| MIL-88C (Cr) | $[Cr_3(\mu_3\text{-}O)(COO)_6]$ | 2,6-NDC | NA | [94] |
| MIL-88D (Cr) | $[Cr_3(\mu_3\text{-}O)(COO)_6]$ | BPDC | NA | [94] |
| MIL-96 (Cr) | $[Cr_3(\mu_3\text{-}O)(COO)_6]$ $[Cr(OH)(COO)_2]_n$ | BTC | NA | [95] |
| MIL-100 (Cr) | $[Cr_3(\mu_3\text{-}O)(COO)_6]$ | BTC | 3100 [c] | [96] |
| MIL-101 (Cr) | $[Cr_3(\mu_3\text{-}O)(COO)_6]$ | BDC | 4100 | [63] |
| MIL-101-NDC (Cr) | $[Cr_3(\mu_3\text{-}O)(COO)_6]$ | 2,6-NDC | 2100 | [97] |
| PCN-333 (Cr) | $[Cr_3(\mu_3\text{-}O)(COO)_6]$ | TATB | 2548 | [98] |
| PCN-426 (Cr) | $[Cr_3(\mu_3\text{-}O)(COO)_6]$ | TMQPTC | 3155 | [99] |
| MIL-53 (Fe) | $[Fe(OH)(COO)_2]_n$ | BDC | NA | [100] |
| MIL-68 (Fe) | $[Fe(OH)(COO)_2]_n$ | BDC | 665 | [101] |
| MIL-141 (Fe) | $[Fe(OH)(COO)_2]_n$ | TCPP | 420 | [102] |
| FepzTCPP (FeOH)2 | $[Fe(OH)(COO)_2]_n$ | Pyrazine, TCPP | 760 | [102] |
| MIL-88A (Fe) | $[Fe_3(\mu_3\text{-}O)(COO)_6]$ | FUM | NA | [94] |
| MIL-88B (Fe) | $[Fe_3(\mu_3\text{-}O)(COO)_6]$ | BDC | NA | [94] |
| MIL-88C (Fe) | $[Fe_3(\mu_3\text{-}O)(COO)_6]$ | 2,6-NDC | NA | [94] |
| MIL-88D (Fe) | $[Fe_3(\mu_3\text{-}O)(COO)_6]$ | BPDC | NA | [94] |
| MIL-100 (Fe) | $[Fe_3(\mu_3\text{-}O)(COO)_6]$ | BTC | 2800 [c] | [103] |
| MIL-101 (Fe) | $[Fe_3(\mu_3\text{-}O)(COO)_6]$ | BDC | 2823 | [104] |
| PCN-250 (Fe) | $[Fe_3(\mu_3\text{-}O)(COO)6]$ | ABDC | 1486 | [105] |
| PCN-250 (Fe$_2$Co) | $[Fe_2Co(\mu_3\text{-}O)(COO)_6]$ | ABDC | 1400 | [105] |
| PCN-333 (Fe) | $[Fe_3(\mu_3\text{-}O)(COO)_6]$ | TATB | 2427 | [90] |
| PCN-600 (Fe) | $[Fe_3(\mu3\text{-}O)(COO)_6]$ | TCPP | 2270 | [106] |
| Tb$_2$(BDC)$_3$ | $[Tb(H_2O)_2(COO)_3]_n$ | BDC | NA | [107] |
| MIL-63 | $[Eu_2(\mu_3\text{-}OH)_7(COO)]_n$ | BTC | 15 | [108] |
| MIL-83 | $[Eu(\mu_3\text{-}O)_3(COO)_3(COOH)_3]_n$ | 1,3-ADC | NA | [109] |
| MIL-103 | $[Tb(H_2O)(COO)_4]_n$ | BTB | 930 | [110] |

| | | | | |
|---|---|---|---|---|
| Y-BTC | $[Y(H_2O)(COO)_3]n$ | BTC | 1080 | [111] |
| Tb-BTC | $[Tb(H_2O)(COO)_3]n$ | BTC | 786 | [111] |
| Y-FTZB | $[Y_6(\mu_3\text{-}OH)_8(COO)_6(CN_4)_6]$ | FTZB | 1310 | [112] |
| Tb-FTZB | $[Tb_6(\mu_3\text{-}OH)_8(COO)_6(CN_4)_6]$ | FTZB | 1220 | [112] |
| Y-FUM | $[Y_6(\mu_3\text{-}OH)_8(COO)_{12}]$ | FUM | 691 | [113] |
| Tb-FUM | $[Tb_6(\mu3\text{-}OH)8(COO)_{12}]$ | FUM | 503 | [113] |
| Ce-UiO-66 | $[Ce_6(\mu_3\text{-}O)_4(\mu_3\text{-}OH)_4(COO)_{12}]$ | BDC | 1282 | [114] |
| Ce-UiO-66-(CH3)$_2$ | $[Ce_6(\mu_3\text{-}O)_4(\mu_3\text{-}OH)_4(COO)_{12}]$ | BDC-$(CH_3)_2$ | 845 | [115] |
| MIL-125 | $[Ti_8O_8(OH)_4(COO)_{12}]$ | BDC | 1550 | [116] |
| PCN-22 | $[Ti_7O_6(COO)_{12}]$ | TCPP | 1284 | [117] |
| COK-69 | $[Ti_3O_3(COO)_6]$ | CDC | NA | [118] |
| MOF-901 | $[Ti_6O_6(OMe)_6(COO)_6]$ | AB, BDA | 550 | [119] |
| MOF-902 | $[Ti_6O_6(OMe)_6(COO)_6]$ | AB, BPDA | 400 | [120] |
| UiO-66 | $[Zr_6(\mu_3\text{-}O)_4(\mu_3\text{-}OH)_4(COO)_{12}]$ | BDC | 1187 | [121] |
| UiO-67 | $[Zr_6(\mu_3\text{-}O)_4(\mu_3\text{-}OH)_4(COO)_{12}]$ | BPDC | 3000 | [121] |
| UiO-68 | $[Zr_6(\mu_3\text{-}O)_4(\mu_3\text{-}OH)_4(COO)_{12}]$ | TPDC | 4170 | [121] |
| PCN-94 | $[Zr_6(\mu_3\text{-}O)_4(\mu_3\text{-}OH)_4(COO)_{12}]$ | ETTC | 3377 | [122] |
| PCN-222 | $[Zr_6(\mu_3\text{-}O)_4(\mu_3\text{-}OH)_4(OH)_4(H_2O)_4(COO)_8]$ | TCPP | 2223 | [123] |
| PCN-223 | $[Zr_6(\mu_3\text{-}O)_4(\mu_3\text{-}OH)_4(COO)_{12}]$ | TCPP | 1600 | [124] |
| PCN-224 | $[Zr_6(\mu_3\text{-}O)_4(\mu_3\text{-}OH)_4(OH)_6(H_2O)_6(COO)_6]$ | TCPP | 2600 | [125] |
| PCN-225 | $[Zr_6(\mu_3\text{-}O)_4(\mu_3\text{-}OH)_4(OH)_4(H_2O)_4(COO)_8]$ | TCPP | 1902 | [126] |
| PCN-228 | $[Zr_6(\mu_3\text{-}O)_4(\mu_3\text{-}OH)_4(COO)_{12}]$ | TCP-1 | 4510 | [127] |
| PCN-229 | $[Zr_6(\mu_3\text{-}O)_4(\mu_3\text{-}OH)_4(COO)_{12}]$ | TCP-2 | 4619 | [127] |
| PCN-230 | $[Zr_6(\mu_3\text{-}O)_4(\mu_3\text{-}OH)_4(COO)_{12}]$ | TCP-3 | 4455 | [127] |
| PCN-521 | $[Zr_6(\mu_3\text{-}O)_4(\mu_3\text{-}OH)_4(OH)_4(H_2O)_4(COO)_8]$ | MTBC | 3411 | [128] |
| PCN-700 | $[Zr_6(\mu_3\text{-}O)_4(\mu_3\text{-}OH)_4(OH)_4(H_2O)_4(COO)_8]$ | Me2BPDC | 1807 | [129] |
| PCN-777 | $[Zr_6(\mu_3\text{-}O)_4(\mu_3\text{-}OH)_4(OH)_6(H_2O)_6(COO)_6]$ | TATB | 2008 | [130] |
| PCN-133 | $[Zr_6(\mu_3\text{-}O)_4(\mu_3\text{-}OH)_4(COO)_{12}]$ | BTB, DCDPS | 1462 | [131] |
| PCN-134 | $[Zr_6(\mu_3\text{-}O)_4(\mu_3\text{-}OH)_4(OH)_2(H_2O)_2(COO)10]$ | BTB, TCPP | 1946 | [131] |
| MOF-801 | $[Zr_6(\mu_3\text{-}O)_4(\mu_3\text{-}OH)_4(COO)_{12}]$ | FUM | 990 | [132] |
| MOF-802 | $[Zr_6(\mu_3\text{-}O)_4(\mu_3\text{-}OH)_4(OH)_2(H_2O)_2(COO)10]$ | PZDC | NA | [132] |
| MOF-808 | $[Zr_6(\mu_3\text{-}O)_4(\mu_3\text{-}OH)_4(OH)_6(H_2O)_6(COO)_6]$ | BTC | 2060 | [132] |
| MOF-812 | $[Zr_6(\mu_3\text{-}O)_4(\mu_3\text{-}OH)_4(COO)_{12}]$ | MTB | 2335 | [132] |
| MOF-841 | $[Zr_6(\mu_3\text{-}O)_4(\mu_3\text{-}OH)_4(OH)_4(H_2O)_4(COO)_8]$ | MTB | 1390 | [132] |
| MOF-525 | $[Zr_6(\mu_3\text{-}O)_4(\mu_3\text{-}OH)_4(COO)_{12}]$ | TCPP | 2620 | [133] |
| MOF-535 | $[Zr_6(\mu_3\text{-}O)_4(\mu_3\text{-}OH)_4(COO)_{12}]$ | XF | 1120 | [133] |
| MOF-545 | $[Zr_6(\mu_3\text{-}O)_4(\mu_3\text{-}OH)_4(OH)_4(H_2O)_4(COO)_8]$ | TCPP | 2260 | [133] |
| DUT-51 | $[Zr_6(\mu_3\text{-}O)_4(\mu_3\text{-}OH)_4(OH)_4(H_2O)_4(COO)_8]$ | DTTDC | 2335 | [134] |
| DUT-52 | $[Zr_6(\mu_3\text{-}O)_4(\mu_3\text{-}OH)_4(COO)_{12}]$ | 2,6-NDC | 1399 | [135] |
| DUT-84 | $[Zr_6(\mu_3\text{-}O)_4(\mu_3\text{-}OH)_4(OH)_6(H_2O)_6(COO)_6]$ | 2,6-NDC | 637 | [135] |
| DUT-67 | $[Zr_6(\mu_3\text{-}O)_4(\mu_3\text{-}OH)_4(OH)_4(H_2O)_4(COO)_8]$ | TDC | 1064 | [136] |
| DUT-68 | $[Zr_6(\mu_3\text{-}O)_4(\mu_3\text{-}OH)_4(OH)_4(H_2O)_4(COO)_8]$ | TDC | 891 | [136] |
| DUT-69 | $[Zr_6(\mu_3\text{-}O)_4(\mu_3\text{-}OH)_4(OH)_2(H_2O)_2(COO)10]$ | TDC | 560 | [136] |
| NU-1000 | $[Zr_6(\mu_3\text{-}O)_4(\mu_3\text{-}OH)_4(OH)_4(H_2O)_4(COO)_8]$ | TBAPy | 2320 | [137] |
| NU-1100 | $[Zr_6(\mu_3\text{-}O)_4(\mu_3\text{-}OH)_4(COO)_{12}]$ | PTBA | 4020 | [138] |
| NU-1101 | $[Zr_6(\mu_3\text{-}O)_4(\mu_3\text{-}OH)_4(COO)_{12}]$ | Py-XP | 4422 | [139] |
| NU-1102 | $[Zr_6(\mu_3\text{-}O)_4(\mu_3\text{-}OH)_4(COO)_{12}]$ | Por-PP | 4712 | [139] |
| NU-1103 | $[Zr_6(\mu_3\text{-}O)_4(\mu_3\text{-}OH)_4(COO)_{12}]$ | Py-PTP | 5646 | [139] |
| NU-1104 | $[Zr_6(\mu_3\text{-}O)_4(\mu_3\text{-}OH)_4(COO)_{12}]$ | Por-PTP | 5290 | [139] |
| MIL-140A | $[ZrO(COO)_2]n$ | BDC | 415 | [140] |
| MIL-140B | $[ZrO(COO)_2]n$ | 2,6-NDC | 460 | [140] |
| MIL-140C | $[ZrO(COO)_2]n$ | BPDC | 670 | [140] |
| MIL-140D | $[ZrO(COO)_2]n$ | Cl2ABDC | 701 | [140] |
| BUT-12 | $[Zr_6(\mu_3\text{-}O)_4(\mu_3\text{-}OH)_4(OH)_4(H_2O)_4(COO)_8]$ | CTTA | 3387 | [141] |

| BUT-13 | $[Zr_6(\mu_3\text{-}O)_4(\mu_3\text{-}OH)_4(OH)_4(H_2O)_4(COO)_8]$ | TTNA | 3948 | [141] |
|---|---|---|---|---|
| Zr-ABDC | $[Zr_6(\mu_3\text{-}O)_4(\mu_3\text{-}OH)_4(COO)_{12}]$ | ABDC | 3000 | [142] |
| BUT-30 | $[Zr_6(\mu_3\text{-}O)_4(\mu_3\text{-}OH)_4(COO)_{12}]$ | EDDB | 3940 | [143] |
| PIZOF | $[Zr_6(\mu_3\text{-}O)_4(\mu_3\text{-}OH)_4(COO)_{12}]$ | PEDC | 2080 | [144] |
| Zr-BTDC | $[Zr_6(\mu_3\text{-}O)_4(\mu_3\text{-}OH)_4(COO)_{12}]$ | BTDC | 2207 | [145] |
| Zr-BTBA | $[Zr_6(\mu_3\text{-}O)_4(\mu_3\text{-}OH)_4(COO)_{12}]$ | BTBA | 4342 | [146] |
| Zr-PTBA | $[Zr_6(\mu_3\text{-}O)_4(\mu_3\text{-}OH)_4(COO)_{12}]$ | PTBA | 4116 | [146] |
| Zr-BTB | $[Zr_6(\mu_3\text{-}O)_4(\mu_3\text{-}OH)_4(OH)_6(H_2O)_6(COO)_6]$ | BTB | 613 | [147] |
| **hcp** UiO-67 | $[Hf_{12}(\mu_3\text{-}O)_8(\mu_3\text{-}OH)_8(\mu_2\text{-}OH)_6(COO)_{18}]$ | BPDC | 1424 | [148] |
| Zr$_{12}$-TPDC | $[Zr_{12}(\mu_3\text{-}O)_8(\mu_3\text{-}OH)_8(\mu_2\text{-}OH)_6(COO)_{18}]$ | TPDC | 1967 | [149] |
| Hf$_{12}$-BTE | $[Hf_{12}(\mu_3\text{-}O)_8(\mu_3\text{-}OH)_8(\mu_2\text{-}OH)_6(COO)_{18}]$ | BTE | NA | [150] |
| Cu-BTPP | $[Cu_3(\mu_3\text{-}OH)(PZ)_3]$ | BTPP | 660 | [151] |
| Ni$_3$(BTP)$_2$ | $[Ni_4(PZ)_8]$ | BTP | 1650 | [152] |
| Zn (1,4-BDP) | $[Zn(PZ)_2]n$ | 1,4-BDP | 1710 | [153] |
| Zn (1,3-BDP) | $[Zn(PZ)_2]n$ | 1,3-BDP | 820 | [153] |
| PCN-601 | $[Ni_8(OH)_4(H_2O)_2(PZ)_{12}]$ | TPP | 1309 | [154] |
| ZIF-8 | $[ZnN_4]$ | mIM | 1947 | [155] |
| ZIF-11 | $[ZnN_4]$ | bIM | 1676 | [155] |
| ZIF-67 | $[CoN_4]$ | mIM | 1587 | [156] |
| ZIF-90 | $[ZnN_4]$ | ICA | 1270 | [157] |
| ZIF-68 | $[ZnN_4]$ | nIM, bIM | 1220 | [64] |
| ZIF-69 | $[ZnN_4]$ | nIM, 5cbIM | 1070 | [64] |
| ZIF-70 | $[ZnN_4]$ | IM, nIM | 1970 | [64] |

[a] These MOFs can be modified by functional organic compounds such as amino, nitro, methyl, halogen, or hydroxyl groups. These MOFs are not explained in this paper; [b] All linkers name are abbreviations [68]; [c] Langmuir surface area

## 2.2. Tetravalent Metal-Carboxylate Based MOFs

Tetravalent metals, such as $Ce^{4+}$, $Zr^{4+}$, and $Ti^{4+}$, and carboxylate linker based MOFs are a comparatively new field of study. Lillerud et al. and Férey et al. have reported on Zr-MOFs and Ti-MOFs, respectively [116,121]. Both Zr- and Ti-MOFs have been applied in various fields because of their high stability [68]. On the other hand, Ce-MOFs are fascinating materials due to their redox properties and possible catalytic activity. For example, a Ce-MOF, which is composed of $Ce^{3+}$ and $Ce^{4+}$, exhibits unique oxidase-like catalytic performance [158].

## 2.3. Trivalent Metal-Carboxylate Based MOFs

MOFs that are composed of trivalent metal cations and carboxylate linkers have two main secondary building units (SBUs): (1) The $[M_3(\mu_3\text{-}O)(COO)_6]$ cluster, which includes a $\mu_3$-oxo-centered trimer of $MO_6$ octahedra and (2) the $[M(OH)(COO)_2]n$ chain, which has a $\mu_2$-hydroxo corner sharing $MO_6$ octahedral unit [68].

## 2.4. Divalent Metal-Azolate Based MOFs

Another type of stable MOFs is composed of soft $M^{2+}$ ions and azolate-based ligands while using hard soft acid base (HSAB) theory. Some of the organic reagents are in the form of azolate-based linkers (Table 2) [159]. Azoles generally release a proton to coordinate with the M ions, similar to carboxylic acids [68]. In addition, azoles display well-known coordination properties and the $sp^2$ nitrogen donors in pyridines and azoles are essentially alike, but different to carboxylic acids [68].

**Table 2.** Structure and typical coordination modes of azolates.

| Linker | Structure | Typical Coordination Modes |
|---|---|---|
| Imidazole (HIM) | | |
| Pyrazole (HPZ) | | |
| Triazole (HTZ and HVTZ) | | |
| Tetrazole (HTTZ) | | |

## 3. Toxic Gas Sensors

The detection of toxic gases is important in environmental remediation and human health problems. Accordingly, many groups have studied new sensing materials for latent modification. Schröder et al. have reported several MOFs that are used for the reversible adsorption of nitrogen dioxide [6]. The Dincă group have reported the use of microporous triazolate-based MOFs for the detection of ammonia gas [7]. Salma et al. have studied the synthesis of a foremost chemical sensor for the identification of sulfur dioxide at room temperature (RT) [12].

### 3.1. Robust Porous MOF for Reversible Adsorption of $NO_2$

As one of the major air pollutants, nitrogen dioxide is fatal to the environment and it causes serious health problems [75–78]. Decreasing $NO_x$ contamination is a difficult issue due to the highly active atomic bond with oxygen and corrosive nature [79]. Therefore, various materials, including metal oxides, mesoporous silica, zeolites, and activated carbons, have been studied as $NO_2$ adsorbents. However, these materials show low adsorption capacities and irreversible uptake due to the disproportionation of $NO^+$ and $NO^{3-}$. MOFs have been used as solid adsorbents, but an isothermal adsorption study on $NO_2$ has not been conducted to date. Therefore, Han et al. studied the isothermal adsorption of MOFs and confirmed that MFM-300 (Al) can interact with highly reactive $NO_2$. Consequently, MFM-300 (Al) has great potential as a practical solid absorbent.

Figure 1 shows the adsorption isotherms that were obtained for MFM-300 (Al) in various gases, including $NO_2$, $CO_2$, $SO_2$, CO, $CH_4$, $N_2$, $H_2$, $O_2$, and Ar at room temperature and pressure. The maximum $NO_2$ isotherm uptake was ~4.1 mmol $g^{-1}$ at room temperature and pressure. This value was much higher than modified Y zeolites [160], mixed oxides, such as $Ce_{1-x}Zr_xO_2$ [161], $NH_3$ functionalized SBA-15 [162], urea-modified mesoporous carbons [163], and activated carbons [164]. Furthermore, the crystallinity and sorption capacity were not changed after the cycling of the sorption and desorption steps.

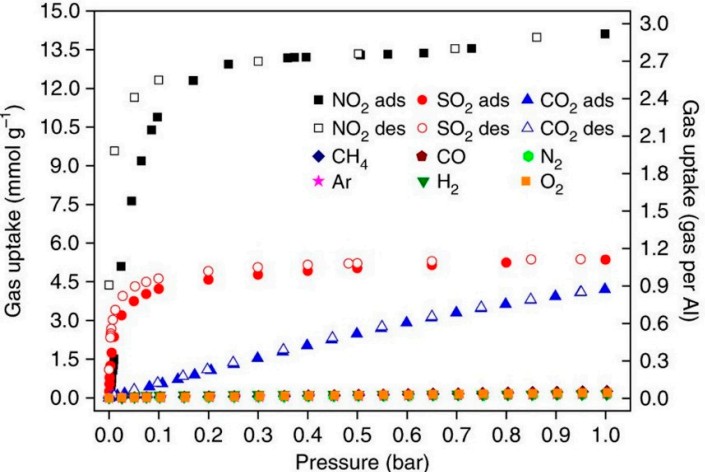

**Figure 1.** Gas uptake properties of MFM-300 (Al) in various gas environment such as $NO_2$, $CO_2$, $SO_2$, $CH_4$, CO, $N_2$, $H_2$, $O_2$, and Ar (open symbol: adsorption, solid symbol: desorption). Reproduced and adapted from Ref. [6]; Copyright (2018), Springer Nature.

### 3.2. Ammonia Sorption in MOFs

Ammonia ($NH_3$) is present in the atmosphere due to global agriculture and industry. The current industrial standard sorbents show a low affinity and limited capacity for $NH_3$. The heterogeneous pore size in carbon-based materials has caused a fundamental problem in studies on ammonia sorption. Recent studies have focused on sorbents containing Lewis or Brønsted acid sites that show a higher affinity toward $NH_3$ molecules to solve this problem. In this study, Dinca et al. showed the static and dynamic ammonia capacities of various microporous triazolate-based MOFs.

Dynamic breakthrough measurements using 0.1% $NH_3$ showed that $Co_2Cl_2BBTA$ and $Co_2Cl_2BTDD$ have a $NH_3$ breakthrough capacity of 8.56 and 4.78 mmol $g^{-1}$, respectively (Table 3). This is equal to 1.48 and 1.08 molecules of $NH_3$ per Co atom, respectively. The $NH_3$ breakthrough capacity value is reduced in 80% relative humidity (RH), regardless of the pore size due to the adsorption of water. The saturation value was 4.36 mol $kg^{-1}$ for $Co_2Cl_2BBTA$ and 3.38 mol $kg^{-1}$ for $Co_2Cl_2BTDD$. These results indicate 0.76 and 0.77 $NH_3$ molecules are absorbed per open metal site in $Co_2Cl_2BBTA$ and $Co_2Cl_2BTDD$, respectively.

**Table 3.** Saturation $NH_3$ breakthrough capacities value at 0.1% of MOFs.

| | Dry (0% RH) | Wet (80% RH) |
|---|---|---|
| $Co_2Cl_2BTDD$ | 4.78 | 3.38 |
| $Co_2Cl_2BBTA$ | 8.56 | 4.36 |
| $Cu_2Cl_2BBTA$ | 7.52 | 5.73 |

### 3.3. Highly Performance of $SO_2$ MOF Sensor

Although sulfur dioxide is one of the most toxic and serious air pollutants, consumption for fossil fuel is increasing [10]. The main adverse health issues occur upon continuous exposure to $SO_2$, with a primary 1-h acceptable limit of 75 ppb. Thus, a sensitive sensor, which can detect even a small amount of $SO_2$ gas, is necessary. However, the detection of $SO_2$ gas by chemical reaction from CaO to $CaSO_3$ has low efficiency and irreversibility [165,166]. Therefore, it is necessary to achieve the reversible physisorption and selective interaction with $SO_2$. Thus far, there are many studies that have been conducted based on the metal oxide ($SnO_2$, $WO_3$, and $TiO_2$) that show high- sensitivity, recover time, and selectivity. However, a sensor based on metal oxide requires the high temperature (200–600 °C), which means that it requires high energy and power.

Recently, MOFs are attractive because of satisfying these requirements mentioned above. However, one of the issues using MOFs in sensing devices is directly related to the fabrication as thin films form. In this study, Salama et al. showed the fabrication of a MOF thin film on various supports and its gas-sensing properties.

Among the various MOFs, such as MFM-300 (Al), MFM-202-a, $Zn_3[Co(CN)_6]_2$, $Co_3[Co(CN)_6]_2$, Mg-MOF-74, and $Ni(bdc)(ted)_{0.5}$, they have chosen the MFM-300 (In) MOF because of its high sorption capacity. To confirm the sensing properties of MFM-300 (In) and measured the changing the capacitance. In particular, MFM-30 (In) MOF was grown on a prefunctionallized IDE with an OH-terminated self-assembled monolayer (SAM) under optimized conditions [167]. X-ray diffraction (XRD) and scanning electron microscopy (SEM) confirmed the structural properties of the film.

The MFM-300 (In) MOF sensor showed exceptional performance. In addition, it can detect $SO_2$ in the ppb range down to 75 ppb (Figure 2). The remarkable detection properties are related to the change in the permittivity of the thin film, depending on the adsorption of $SO_2$ molecules. There are two types of interaction in the adsorption process: (1) Analyte-framework interactions and (2) analyte-analyte interactions. Changing these two interactions induce a change in the capacitance of the thin film. The MOF sensor exhibited good stability over three weeks of operation.

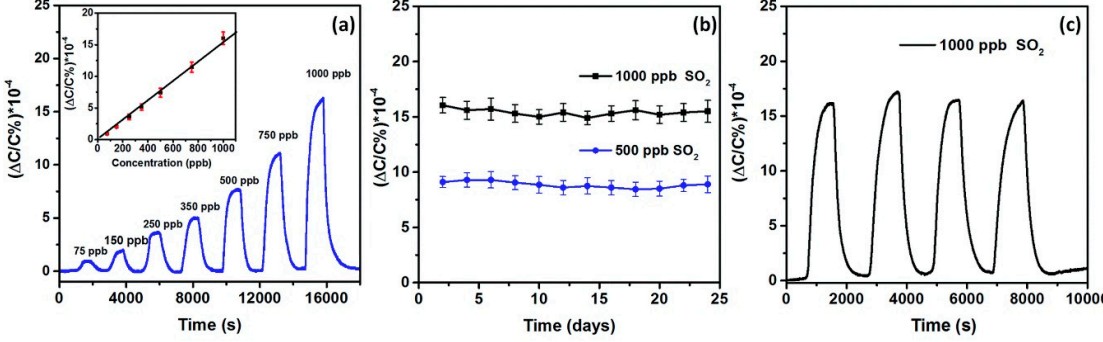

**Figure 2.** (**a**) Detection of $SO_2$ in the 75 to 1000 ppb concentration range (**b**) linear response for MFM-300 (In) MOF-sensor upon exposure to $SO_2$ for a 24-day (**c**) reproducibility cycles for the detection. Reproduced and adapted from Ref. [12]; Copyright (2018), RSC Journal of Materials Chemistry A.

They also measured the sensing performance with relative humidity (RH) at 350 and 1000 ppb of $SO_2$ gas. However, it does not show distinctive signals as compared to the "dry" condition (Figure 3a). Therefore, it confirmed that the practical applicability of MFM-300 (in) MOF as $SO_2$ sensor in

humidity condition. The temperature dependent sensing properties from 22 °C to 80 °C showed that the performance decrease up to 35% with increasing temperature (Figure 3b). Finally, selectivity performance was conducted in various gases of MFM-300 (In) MOF. As a result, it showed good selectivity for $SO_2$ gas when compared to others (Figure 3c).

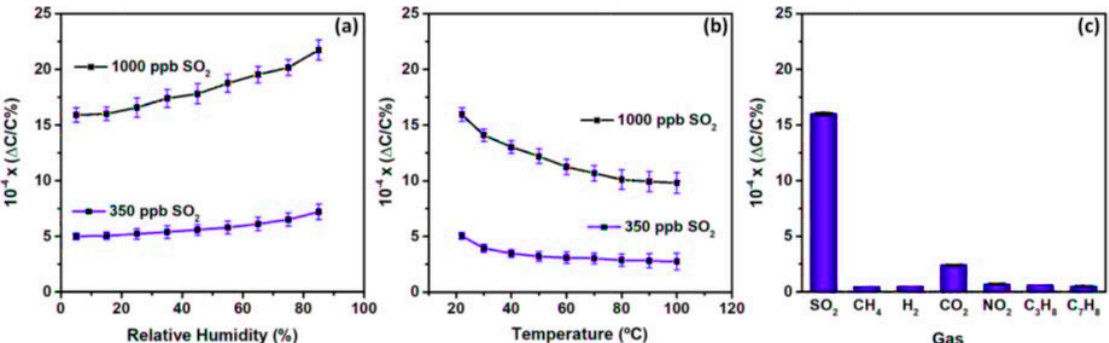

**Figure 3.** Environment effect of sensing performance with (**a**) relative humidity (**b**) temperature and (**c**) selectivity performance in various gases of MFM-300 (In) MOF sensor. Reproduced and adapted from Ref. [12]; Copyright (2018), RSC Journal of Materials Chemistry A.

## 4. Detection and Reduction of Toxic Water via MOFs

### 4.1. Detection of Toxic 4-Nitrophenol via AgPd Nanoparticles on Functionalized MOFs

Nitroaromatic compounds are continuous organic contaminants that originate from industrial waste. 4-Nitrophenol (4-NP) is one of the harmful phenolic pollutants found in chemical waste [54], which is due to its high polarity and subsequent high solubility in water.

4.1.1. Synthesis of UiO-66-L and AgPd Nanoparticles Embedded on UiO-66-L (L=$NH_2$ and $NO_2$)

The synthesis of UiO-66-L was previously reported in the literature [168]. $ZrCl_4$ was dispersed in DMF and the resulting mixture activated with acetic acid at 55 °C. $2-NH_2-H_2BDC$ and $2-NO_2-H_2BDC$ were used to functionalize UiO-66, respectively.

AgPd@UiO-66-L MOF was prepared via a reduction method while using sodium borohydride. UiO-66-L MOFs were homogeneously dispersed in water, and $AgNO_3$ and $PdCl_2$ were dispersed in the resulting MOFs dispersion, respectively. An aqueous solution of $NaBH_4$ was added to the $Ag^+$, $Pd^{2+}$, and MOFs mixture to reduce the Ag and Pd. The product was isolated via centrifugation, washing, and drying (Scheme 2).

Scheme 2. Synthesis of AgPd@UiO-66-L and the electrochemical reduction mechanism of 4-nitrophenol.

### 4.1.2. Characterization of UiO-66-L and AgPd@UiO-66-L

In this review, the characterization of UiO-66-L and AgPd@UiO-66-L while using SEM, TEM, and XRD is described. The FE-SEM and TEM images show the good dispersion of metal NPs on the AgPd@UiO-66-NH$_2$ MOF and its octahedral morphology (Figure 4a,b). The elemental distribution of UiO-66-NH$_2$ was investigated while using HAADF and elemental mapping (Figure 4c–j), which confirmed the bimetallic AgPd nanoparticles were loaded into both the bulk MOF and on its surface.

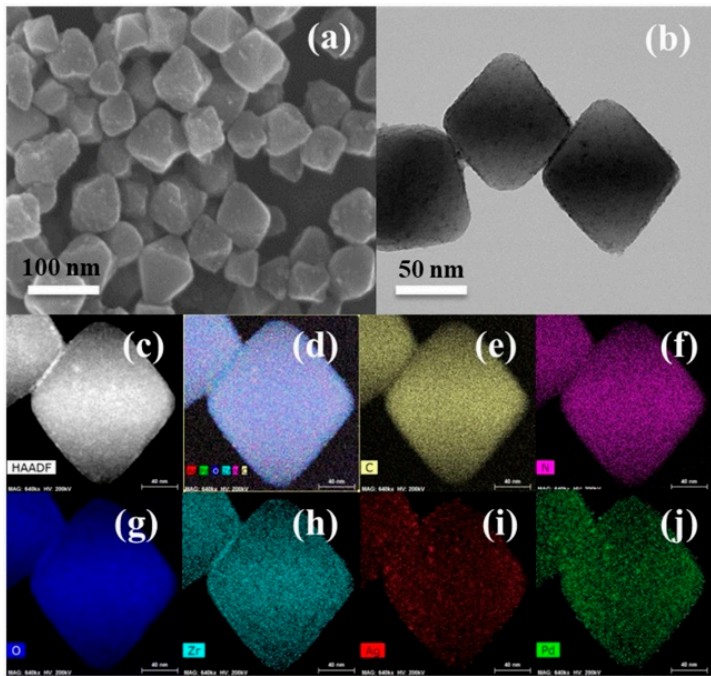

**Figure 4.** Scanning electron microscopy (SEM) image (**a**), TEM image (**b**), HAADF image (**c**), total element (**d**), and elemental mapping (**e–j**) of AgPd@UiO-66-NH$_2$. The bar indicate 100 nm (**a**), 50 nm (**b**), and 40 nm (**c–j**). Reproduced and adapted from Ref. [11]; Copyright (2018), Elsevier Sensors and Actuators B: Chemical.

The XRD spectra showed the crystallinity and structural aspects of the as-synthesized materials (Figure 5). The XRD spectra that were recorded for UiO-66-NH$_2$ and UiO-66-NO$_2$ were similar and in good agreement with AgPd and bare UiO-66 (Figure 5a–c). This shows that the materials crystallinity was not changed after functionalization of the -NH$_2$ and -NO$_2$ groups (Figure 5d,e). The characteristic peaks for Ag and Pd were observed at 2θ = 38.03° and 40.01°, respectively. Moreover, the existence of both Ag and Pd peaks in Figure 5f,g, respectively, show the obvious crystallinity of the AgPd alloy.

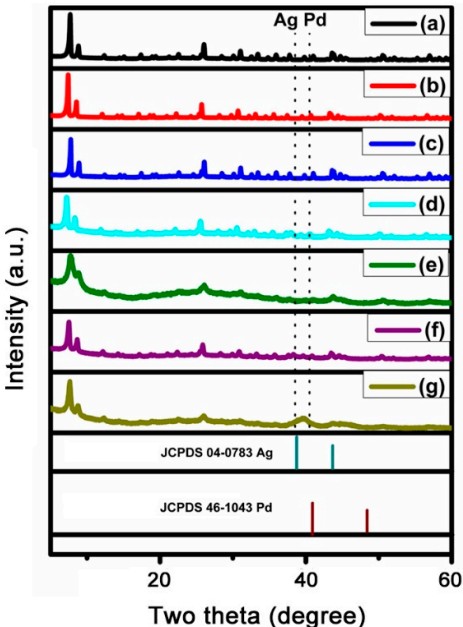

**Figure 5.** PXRD pattern of UiO-66 (**a**), UiO-66-NH₂ (**b**), UiO-66-NO₂ (**c**), Ag@UiO-66-NH₂ (**d**), Pd@UiO-66-NH₂ (**e**), AgPd@UiO-66-NH₂ (**f**), and AgPd@UiO-66-NO₂ (**g**). Reproduced and adapted from Ref. [11]; Copyright (2018), Elsevier Sensors and Actuators B: Chemical.

### 4.1.3. A Comparison of the Catalytic Performance by the Detection and Reduction of 4-Nitrophenol

In this paper, 0.5 mM of 4-nitrophenol was detected while using AgPd@UiO-66-L by cyclic voltammetry. The sharp reduction peak at –0.7 V vs. Ag/AgCl shows the reduction of 4-nitrophenol to 4-hydroxyaminophenol by AgPd@UiO-66-NH₂/GCE (Figure 6a). In addition, a quasi-reversible anodic peak was observed at ~0.1 V vs. Ag/AgCl, due to the oxidation of 4-hydroxyaminophenol to 4-nitrosophenol. In this result, AgPd@UiO-66-NH₂ showed improved performance when compared to the other materials studied. Figure 6b shows the cyclic voltammograms for the sensing of various concentrations of 4-nitrophenol (0.25–400 μm) by the AgPd@UiO-66-NH₂/GCE electrode in 0.1 M phosphate buffered saline (PBS) at pH 7. Increasing the concentration of 4-nitrophenol from 0.25 μM to 400 μM exhibited a linear increase in the cathodic reduction peak due to the beneficial electrocatalytic reduction of 4-nitrophenol to 4-hydroxyaminophenol.

The catalytic activities of the as-synthesized products were also studied while using this catalytic reduction reaction using UV-Vis spectroscopy. 4-Nitrophenol shows an absorption peak at ~317 nm in an aqueous medium. Upon the addition of NaBH₄ powder to the solution, a new peak was detected at 400 nm (Figure 6c). The resulting thick yellow mixture and shift in the absorption peak was due to the generation of p-nitrophenolate [169]. Figure 6d showed catalytic activity by AgPd@UiO-66-NH₂ in the mixed solution of sodium borohydride and 4-nitrophenol. The color of mixed solution altered from yellow to colourless because of the activation of 4-nitrophenolate ion to 4-aminophenol.

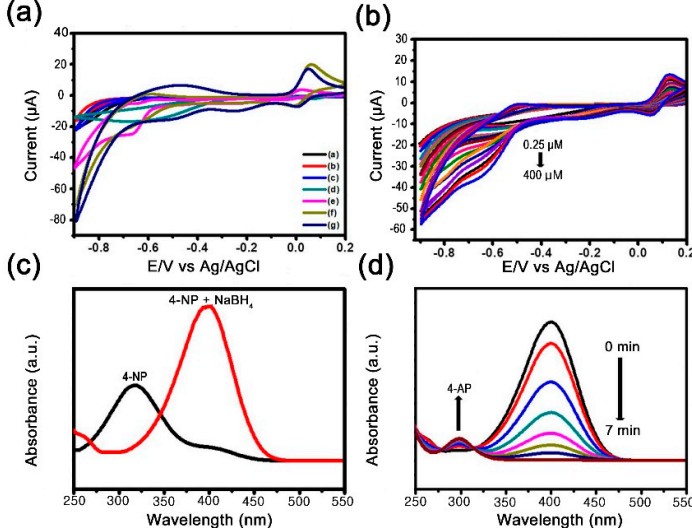

**Figure 6.** (**a**) CVs of the identical electrodes in 0.1 M PBS solution including 0.5 mM of 4-nitrophenol, (**b**) CVs of AgPd@UiO-66-NH$_2$/GCE in 0.1 M PBS at various concentration 4-nitrophenol (0.25 μM– 400 μM), (**c**) UV-VIS absorption spectra of 4-nitrophenol before addition of NaBH$_4$ solution as compared to after, and (**d**) Intensity of absorbance peak change detection via UV-VIS spectra for the reduction of 4-nitrophenol within NaBH$_4$ in AgPd@UiO-66-NH$_2$ during the time-dependent. Reproduced and adapted from Ref. [11]; Copyright (2018), Elsevier Sensors and Actuators B: Chemical.

## 4.2. Detection of Antibiotics in Water Using Cd-MOF as a Fluorescent Probe

Antibiotics are used to treat bacterial infections in animals and humans, but they are an important organic pollutant [170,171]. The abuse of antibiotics has led to extreme residues in subsoil water and surface water [172–177]. Therefore, luminescent MOFs have been synthesized and applied in the detection of antibiotics in water as an alternative to liquid chromatography (LC) combined with UV-Vis spectroscopy, capillary electrophoresis, Raman spectroscopy, mass spectroscopy, and ion mobility spectroscopy [141,178–193]. In this paper, the reported Cd-MOF material was used toward the detection of the antibiotic, ceftriaxone sodium (CRO).

### 4.2.1. Synthesis and Detection of Cd-MOF

The synthesis of Cd-MOF while using 1,4-bis(2-methyl-imidazole-1-yl)butane (bbi) was carried out while using a literature process [194]. The detection of antibiotics was achieved using UV spectroscopy in the wavelength range of 270–350 nm.

### 4.2.2. Characterization of Cd-MOF

The Cd-MOF was analyzed in terms of its thermal and chemical stability under harsh conditions. Figure 7a exhibited no weight loss in the temperature range of 35–300 °C due to the absence of water in the Cd-MOF. The framework started to decompose at temperatures >300 °C. Meanwhile, the Cd-MOF was stored in aqueous solutions of the antibiotic and distilled water under various pH conditions for 24 h (Figure 7b). The PXRD spectra of the treated samples are in good agreement with the base simulated data. In these results, the Cd-MOF exhibited high physical and chemical stability in alkaline, acidic, and antibiotic solutions, respectively.

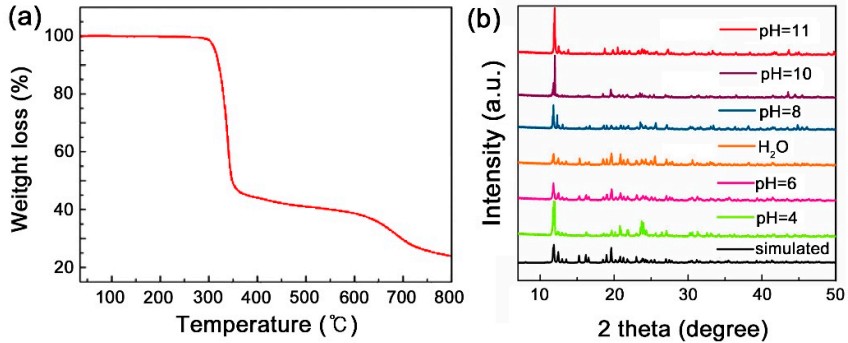

**Figure 7.** (**a**) Thermal gravity analysis graph of Cd-MOF and (**b**) PXRD spectrum of Cd-MOF neglected in various pH aqueous solution and water for 24 h. Reproduced and adapted from Ref. [54]; Copyright (2019), RSC Analyst.

The MOF candidates used as luminescent materials are often constructed from conjugated organic ligand linkers and $d^{10}$ metal ions [195–200]. Therefore, the Cd-MOF shows enhanced fluorescence intensity when compared to those that are constructed from organic ligands, such as bbi and $H_2L$ (Figure 8).

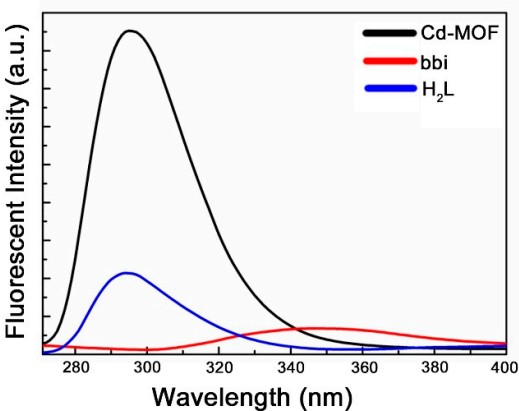

**Figure 8.** The fluorescence spectra of solid samples of Cd-MOF, bbi, and $H_2L$ organic ligand. Reproduced and adapted from Ref. [54]; Copyright (2019), RSC Analyst.

### 4.2.3. Chemical Sensors for Antibiotic Detection

The application of Cd-MOF as a fluorescent sensor for detecting antibiotics in water was investigated because of its robust luminescence properties in water. Cd-MOF was dispersed in antibiotic solutions containing LIN, AZL, PEN, AMK, ERY, AMX, AZM, GEN, ATM, FOX, CSU, CEC, CED, CFM, MTR, SXT, and CRO (LIN: Lincomycin hydrochoride, AZL: Azithromycin latobionate, PEN: Penicillin, AMK: Amikacin, ERY: Erythromycin ethylsuccinate, AMX: Amoxicillin, AZM: Azithromycin, GEN: Gentamicin, ATM: Aztreonam, FOX: Cefoxitin, CSU: Cefathiamidine, CEC: Cefaclor, CED: Cefradine, CFM: Cefixime, MTR: metronidazole, SXT: Sulfamethoxazole, and CRO: Ceftriaxone sodium, respectively). As a result, the effective fluorescence quenching of Cd-MOF by these antibiotics follows the order of AZL < GRN < LIN < AMK < ERY < PEN < AZM < AMX < FOX < CEC < CSU < MTR < CFM < ATM < SXT < CRO (Figure 9a). In particular, the efficient detection of CRO via fluorescence quenching was ~90% in this study. Figure 9b shows the distinction of the Cd-MOF sensor for quantitative analysis while using a fluorescence titration experiment. This graph shows the straightforward and dramatic trend in the fluorescence intensity of Cd-MOF that was detected from 0 to 70 μL. Cd-MOF is an effective probe used to detect CRO. The Cd-MOF sensor in an aqueous solution of CRO exhibits highly sensitive fluorescence intensity

under various pH conditions (Figure 9c, pH = 4–11). From this graph, Cd-MOF can be used under various pH conditions with no effect on the experimental results. In particular, Cd-MOF shows high performance and stability at pH = 6–7. Figure 9d shows that the rapid detection of various concentrations of CRO can be achieved when an aqueous solution of CRO was added to the Cd-MOF suspension.

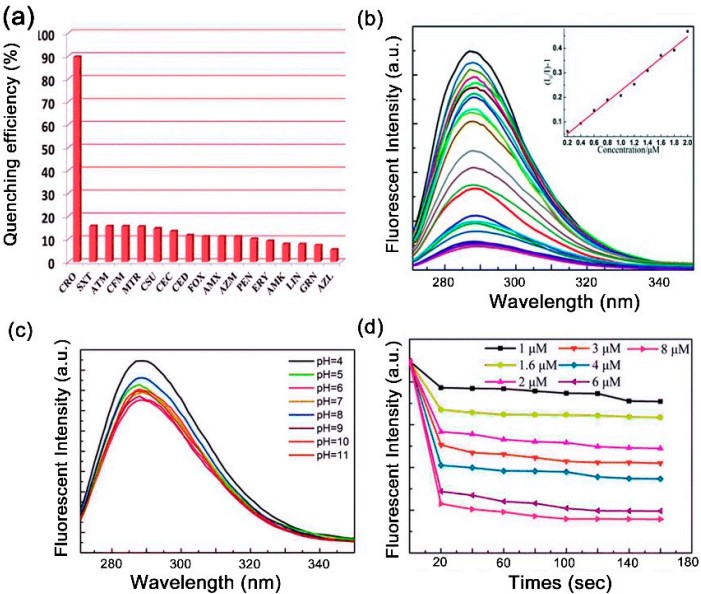

**Figure 9.** (**a**) Comparison of various antibiotics quenching efficiency using Cd-MOF at room temperature. (**b**) Dispersed in aqueous solution of CRO for fluorescence spectra (**c**) CRO aqueous solution tested in various pH aqueous solution using Cd-MOF, and (**d**) Fluorescence intensity of CRO solution in Cd-MOF suspensions in time dependent. Reproduced and adapted from Ref. [54]; Copyright (2019), RSC Analyst.

## 5. Filtration of Water and Air Pollutants Using Nanofibrous MOFs Prepared via Electrospinning Methods

### 5.1. Nanofiber MOF Filter for Particulate Matter

Serious threats to human health have been arisen due to the rapid development of the economy and industry, with the most dangerous of them being air pollution [201,202]. In practice, air pollutants are very diverse and are typically composed of particulate matter (PM) and toxic gases. PM is harmful to the environment affecting human health, air quality and the climate. Particulate matter whose aerodynamic diameters are <2.5 μm (PM2.5) and <10 μm (PM10) can penetrate the respiratory system and cause health problems upon prolonged exposure [203,204]. A lot of attention has been focused on researching PM filters to solve these problems. Among them, research on filters made using MOFs via electrospinning methods is a major factor.

#### 5.1.1. Basic Theory of Electrospinning

Electrospinning is the most facile way to make nanofiber membranes containing organic and inorganic components while using polymer melts and solutions [205]. The basic principle of electrospinning is to apply a strong electric field using a high voltage power supply and drawing the fabricated fiber as they solidify (Scheme 3) [206]. It is easy to install and produce, so this method enables mass production at a low cost. Although the basic principle of electrospinning is simple, its mechanism is very complicated, because there are many factors that affect the process. Among them, the process parameters include the nozzle diameter, applied voltage, and tip-to-collector distance (TCD) [207]. In addition, it is also influenced by the solution characteristics and physical

properties, such as the concentration, viscosity, surface tension, and vapor pressure. In particular, there are many advantages that affect performance of filter, such as specific surface area, alterable fiber diameter, and pore size.

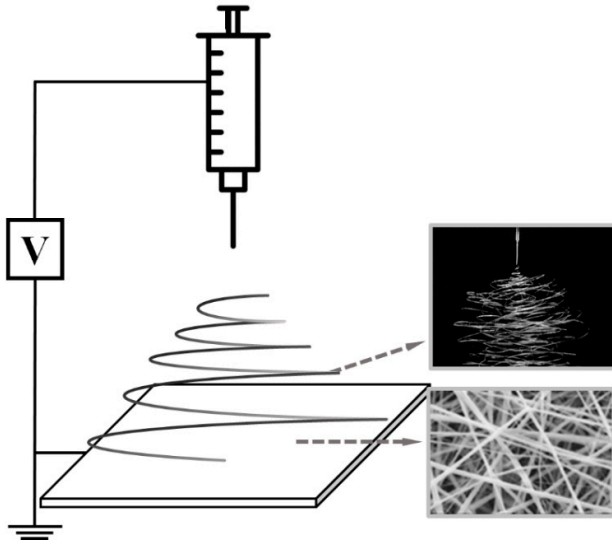

**Scheme 3.** Electrospinning Setup.

### 5.1.2. Detail Mechanism of Electrospinning

Electrospinning is the simplest way for producing micro or nanoscale fibers [208–210]. Although the basic theory of this process is uncomplicated, its detail mechanism is so complicated. Many conditions, such as solution terms and experimental environment, affect the electrospinning process.

In the case of solution term, viscosity, surface tension, vapor pressure and solution conductivity have a major influence on electrospinning. The most important of these is viscosity, which can vary greatly with electrospinning. If all conditions are the same, except for the viscosity, it will affect the thickness and formation of the fibers. If the viscosity of the solution is too high, since the drop for electrospinning is not formed, electrospinning itself might be disrupted. On the other hand, if it is considerably low, the fibers are not able to withstand the tension caused by stretching in the process of drawing fibers, so that the fiber are broken. At the optimum viscosity for electrospinning, generally, if the viscosity increases, the stretching proceeds relatively slowly, and the fibers may become thick and vice versa. Similarly, vapor pressure also affects the thickness of the fiber. The vapor pressure of the solution has an intuitive effect on the evaporation rate of the interpolation solvent, so that adjusting the vapor pressure can control the thickness of the fiber.

Surface tension is thought to influence jet formation after drop formation. If the surface tension is too high, drops are formed, but it is difficult to form a jet. This is because the surface tension of the solution is higher than the high voltage applied and, thus, the jet is not formed. In general, it is a good idea to consider the surface tension of the solution if a drop is formed but no jet is formed.

Electrospinning is basically a process for very low conductivity solutions. Electrospinning on conductive solutions is a very difficult process. The reason is that when the conductivity of the solution is high, the drop itself is not formed, and charge is directly discharged from the solution to the collector, which makes it difficult to form electrospinning. It might be possible to spinning a conductive solution by applying a very high voltage, so that the amount of charge applied is greater than that discharged, in order to spinning a conductive solution. The experimental environmental factors affecting electrospinning include TCD, voltage, nozzle size, and temperature and humidity. In the case of temperature and humidity, it is difficult to control these, which is one of the main reasons why the reproducibility of electrospinning result is different every day. Conditions that are

related to electrospinning settings, such as TCD voltage and nozzle size, have a relatively small impact. During the electrospinning experiment, the TCD and voltage can be changed in real time, so it is relatively easy to know the optimal conditions for spinning. In this context, the detailed mechanisms and conditions of electrospinning are very diverse and complex, but, if the optimum conditions are found, then a fiber that has high propulsion can be obtained.

### 5.1.3. Characterization of MOF@PAN, PS (Polystyrene)

A high ratio of MOFs can be loaded into polymer composites without agglomeration by adjusting the morphology and particle size of dissimilar MOF crystals. Figure 10 shows the SEM images of polymer and various MOF non-woven fabrics. By controlling the electrospinning conditions, such as voltage, flow rate of the solution, and TCD, four MOFs can be formed into fiber materials. Figure 10 shows that the MOF nanoparticles are well dispersed in the polymer fibers without any apparent agglomeration, despite a high loading of 60 wt%.

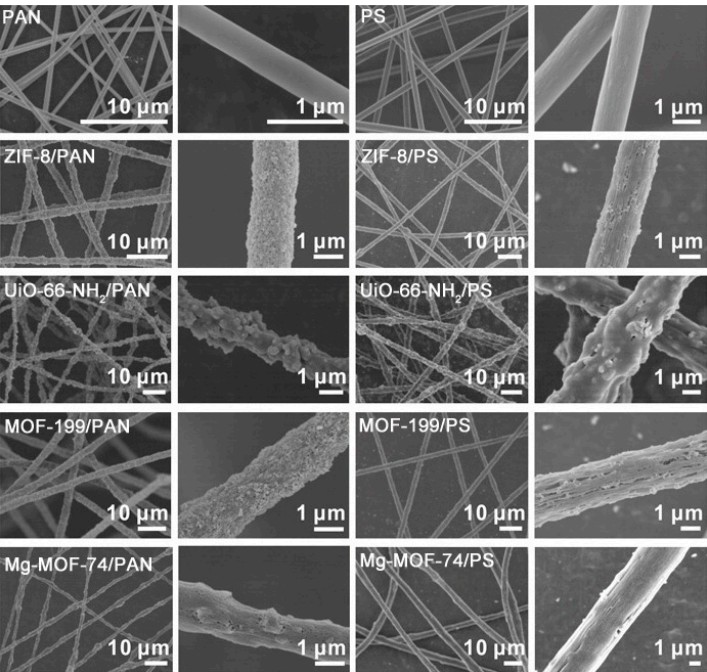

**Figure 10.** SEM images of polymer and various MOFs loaded nonwoven fabrics. Reproduced and adapted from Ref. [58]; Copyright (2016), ACS Journal of American Society.

### 5.1.4. Filtration of Particle Matter Using ZIF-8@PAN

Generally, MOF filters can capture pollutants, including particulate matter via three mechanisms: (1) Pollutants can be bound to the OMSs, (2) interaction with the functional groups in the MOF filters, and (3) electrostatic interactions with the MOF filter. Figure 11 shows the particulate matter removal efficiency that was observed for MOFs@PAN filters. Figure 11a shows that the ZIF-8@PAN filter has the highest removal efficiency for $PM_{2.5}$ and $PM_{10}$ among the various MOFs studied. Particulate matter is very polar because of the presence of water vapor and various ions. MOFs, which have unbalanced defects and metal ions on the surface, offer positive charge. This is why the surface of PM can be polarized enhancing the electrostatic interactions. The zeta potential represents these electrostatic interactions. Among the MOFs studied, ZIF-8 exhibited the highest zeta potential of 47.5 mV. In this context, the ZIF-8@PAN filter displayed higher removal efficiency than the other filters studied and its efficiency was maintained after 48 h of exposure to polluted air (Figure 11b).

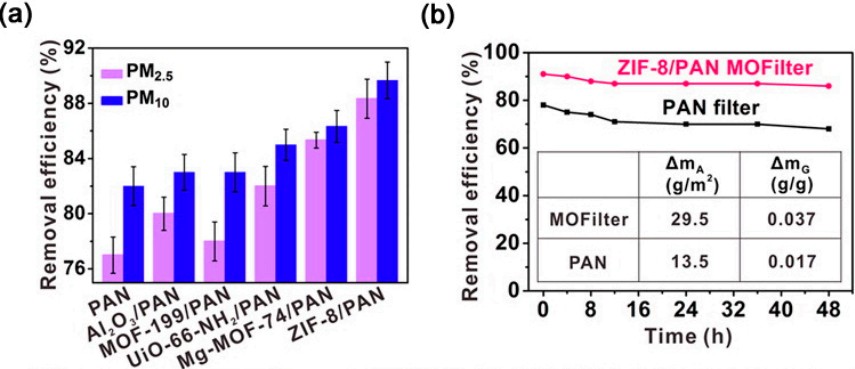

**Figure 11.** (**a**) Particulate matter removal efficiencies of polyacrylonitrile (PAN) filter, Al₂O₃@PAN filter and MOF@PAN filter (**b**) Long term PM2.5 removal efficiencies of PAN filter and ZIF-8@PAN filter. Reproduced and adapted from Ref. [58]; Copyright (2016), ACS Journal of American Society.

### 5.2. Nanofiber MOF Filter for Water Pollutants

Water contamination has become an important issue in environmental remediation due to the increase of urban areas and industrialization. Domestic wastewater is continuously discharged into the environment. Generally, the major pollutants in food wastewater are soluble organic food additives and insoluble organic compounds. Many methods have been used to treat these types of pollutants, such as advanced oxidation, adsorption, and photocatalytic membrane technology [211–215]. Among these methods, membrane technology is preferred due to its facile operation. However, research studies mainly concentrate on the removal of one type of pollutant, either soluble or insoluble pollutants. Recently, porous materials, such as MOFs, have been applied to water filtration to treat these contaminants [211–219]. It is necessary to use an electrospinning process to obtain superhydrophilic-underwater superoleophobic properties. In this context, an electrospun polyacrylonitrile (PAN) and MIL-100 (Fe) composite filter (PAN@ MIL-100(Fe)) have been fabricated to treat domestic wastewater.

### 5.2.1. Schematic of the PAN@MIL-100 (Fe) Filter

Scheme 4 shows the process that is used to prepare the PAN@MIL-100 (Fe) filter. In view of the facile electrospinning process, a H₃BTC/PAN electrospun fiber filter was prepared as the precursor to load MIL-100 (FE), where the PAN fiber is used as a polymer frame and trimesic acid used as the initial reaction site for MIL-100 (FE) growth. As the hydrothermal reaction proceeds, trimesic acid acts as a nucleation site for the growth of MIL-100 (Fe) on the PAN fibers.

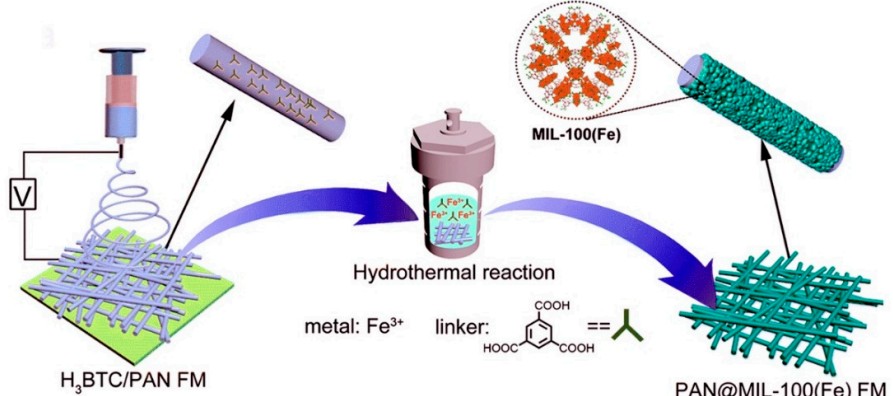

**Scheme 4.** (**a**) Schematic illustration of fabricating the PAN@MIL-100 (Fe) filter. Reproduced and adapted from Ref. [59]; Copyright (2019), RSC Journal of Materials Chemistry A.

### 5.2.2. Characterization of the PAN@MIL-100 (Fe) Filter

Figure 12a and b show that the H₃BTC/PAN fiber filter has a smooth surface without any beads. The average diameter is 110 nm. After the growth process, the PAN@MIL-100 (Fe) filter has a rough fiber surface with lots of particles. Many particles are covered on the PAN fibers with an average diameter of 211 nm, which is increased during the coating process. This indicates that the MOFs are successfully coated onto the PAN fibers without any aggregation.

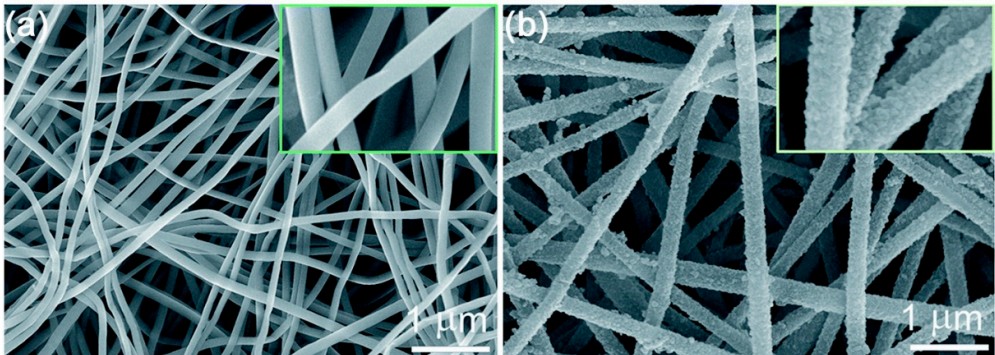

**Figure 12.** SEM image of (**a**) H₃BTC/PAN filter and (**b**) PAN@MIL-100 (Fe) filter. Reproduced and adapted from Ref. [59]; Copyright (2019), RSC Journal of Materials Chemistry A.

### 5.2.3. Filtration of the Wastewater with Soluble Pollutants Using PAN@MIL-100 (Fe) Filter

The electrospun fiber filter that was prepared without MOFs has macro-size pores, so it is difficult to effectively separate the pollutants. The adsorption interactions between the fibers and pollutants is major factor in the filtration performance of the filter. In this context, Figure 13 shows the results of filtering amaranth red (AR) and vanillin (VA) as soluble pollutants. AR and VA are approximately 99% removed and the removal efficiencies were both >95% after 10 adsorption-desorption cycles.

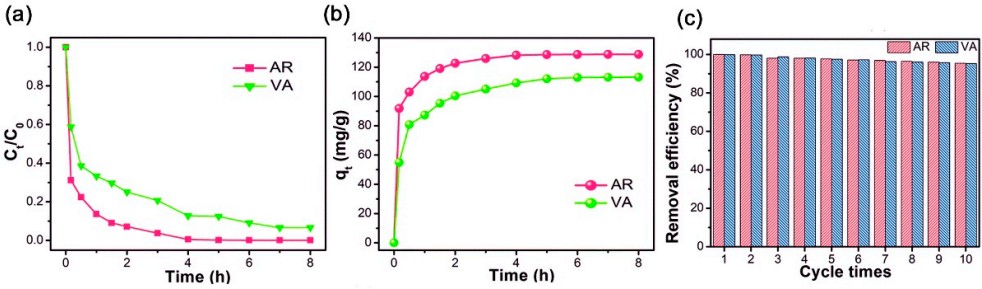

**Figure 13.** (**a**) Removal efficiency and (**b**) adsorption kinetic curves toward AR and VA by PAN@MIL-100 (Fe) filter. (**c**) Adsorption-desorption cycles. Reproduced and adapted from Ref. [59]; Copyright (2019), RSC Journal of Materials Chemistry A.

### 5.2.4. Filtration of Wastewater Containing Insoluble Pollutants Using the PAN@MIL-100 (Fe) Filter

In the removal of oil, an important parameter is the surface wettability. To treat insoluble (oil) pollutants, it is essential that the filter has the property of selective wettability (superhydrophilicity and underwater superoleophobicity), which allows for water to pass through filter, but not oil. The selective wettability is that the filter can wet both water and oil in air, but in water has only hydrophilicity. Its basic mechanism is that water around the filter acts as a barrier to prevent oil from passing through. Figure 14a,b show that the PAN@MIL-100 (Fe) filter has selective wettability and

the underwater oil pollutants contact angles are 151° and 154°. After five cycles, the oil removal efficiency is only slightly changed.

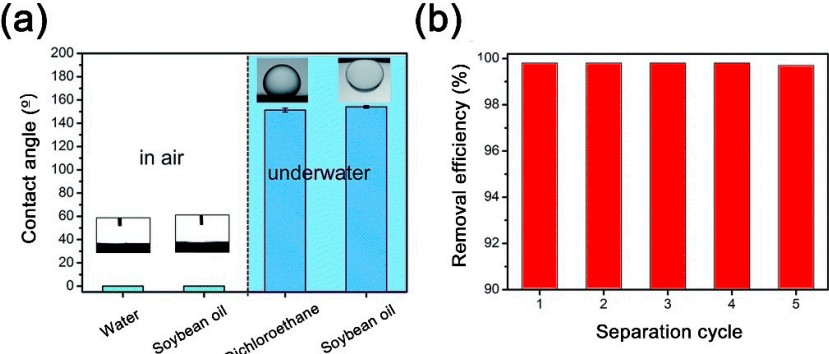

**Figure 14.** (**a**) Contact angles of water and oil in air and under water. (**b**) Separation efficiency versus the recycling number. Reproduced and adapted from Ref. [59]; Copyright (2019), RSC Journal of Materials Chemistry A.

## 6. Conclusions

The application of various MOFs materials was reviewed with a focus on toxic sensor, reduction catalyst, hydrogen storage, and filter, which act as successful functional materials. We reviewed all of the materials while using MOFs that exhibited good performance and various application. The application of MOF material for toxic sensor, such as $NO_2$, $SO_2$, and ammonia gas, showed high performance via MFM-300 (Al), Co based MOFs ($Co_2Cl_2BTDD$ and $Co_2Cl_2BBTA$), and MFM-300 (In). Our previous catalyst e.g., PdAg@UiO-66-L for the detection and reduction of 4-nitrophenol also showed good catalytic activity. Moreover, a new Cd-MOF as a fluorescent detect exhibited high sensitivity and selectivity in CRO. Various MOFs for hydrogen adsorption, such as MOF-177, SNU-6, and MOF-74 composited by Co/Ni mixed-material, exhibited good performance while using physisorption analysis method. Filtration for particle matters and wastewater using organic fiber modified as MOF-based material showed good adsorption in polluted condition. In summary, MOFs can be expected one of the best candidate to solve environmental pollution and energy storage in the near future.

**Author Contributions:** K.H.P. provided academic direction and Discussion of the results and revising the full manuscript. S.J. collected materials and wrote the introduction, basic of stable MOFs, and conclusion part. S.S. and J.H.L. contributed to the materials and methods and results and discussion equally. H.-S.K., T.B.P., K.-T.H., and S.P. participated in discussion of the results. All authors have read and agreed to the published version of the manuscript.

**Funding:** This research was funded by Basic Science Research Program through the National Research Foundation of Korea (NRF) funded by the Ministry of Science, ICT & Future Planning (NRF-2017R1A4A1015533, 2017R1D1A1B03036303 and 2018R1D1A1B07045663). This research was partially supported by VietNam National University Ho Chi Minh City (NCM2019-50-01).

**Acknowledgments:** This research was supported by PNU-RENovation (2018–2019).

**Conflicts of Interest:** The authors declare no conflict of interest.

## Abbreviations

| | |
|---|---|
| BDC | terephthalate |
| FUM | fumarate |
| 2,6-NDC | naphthalene-2,6-dicarboxylate |
| BTC | benzene-1,3,5-tricarboxylate |
| BDC-NH$_2$ | 2-aminoterephthalate |

| | |
|---|---|
| BTEC | 1,2,4,5-benzenetetracarboxylate |
| NTC | 1,4,5,8-naphthalenetetracarboxylate |
| BPDC | biphenyl-4,4′-dicarboxylate |
| BPTA | biphenyl-3,3′,5,5′-tetracarboxylate |
| BTB | 1,3,5-benzenetrisbenzoate |
| BeDC | 4,4′-benzophenonedicarboxylate |
| 1,3-BDC | isophthalate |
| BTTB | 4,4′,4″-[benzene-1,3,5-triyl-tris(oxy)]tribenzoate |
| TCPP | meso-tetrakis(4-carboxylatephenyl)porphyrin |
| TATB | 4,4′,4″-s-triazine-2,4,6-triyl-tribenzoate |
| TCPT | 3,3″,5,5″-tetrakis(4-carboxyphenyl)-p-terphenyl |
| TMQPTC | 2′,3″,5″,6′-tetramethyl-[1,1′:4′,1″:4″,1‴-quaterphenyl]-3,3‴,5,5‴-tetracarboxylate |
| ABDC | 4,4-azobenzenedicarboxylate |
| 1,3-ADC | 1,3-adamantanedicarboxylate |
| FTZB | 2-fluoro-4-(tetrazol-5-yl)benzoate |
| CDC | trans-1,4-cyclohexanedicarboxylate |
| AB | 4-aminobenzoate |
| BDA | benzene-1,4-dialdehyde |
| BPDA | 4,4′-biphenyldicarboxaldehyde |
| TPDC | [1,1′:4′,1″-terphenyl]-4,4″-dicarboxylate |
| ETTC | 4′,4″,4‴,4⁗-(ethene-1,1,2,2-tetrayl)tetrabiphenyl-4-carboxylate |
| MTBC | 4′,4″,4‴,4⁗-methanetetrayltetrabiphenyl-4-carboxylate |
| PZDC | 1H-pyrazole-3,5-dicarboxylate |
| MTB | 4,4′,4″,4‴-methanetetrayltetrabenzoate |
| XF | 4,4′-((1E,1′E)-(2,5-bis((4-carboxylatephenyl)ethynyl)-1,4-phenylene)bis(ethene-2,1-diyl))dibenzoate |
| DTTDC | dithieno[3,2-b;2′,3′-d]-thiophene-2,6- dicarboxylate |
| TDC | 2,5-thiophenedicarboxylate |
| TBAPy | 1,3,6,8-tetrakis(p-benzoate)pyrene |
| PTBA | 4-[2-[3,6,8-tris[2-(4-carboxylatephenyl)-ethynyl]-pyren-1-yl]ethynyl]-benzoate |
| Py-XP | 4′,4‴,4⁗,4‴‴-(pyrene-1,3,6,8-tetrayl) tetrakis(2′,5′-dimethyl-[1,1′-biphenyl]-4-carboxylate |
| Por-PP | meso-tetrakis-(4-carboxylatebiphenyl)- porphyrin |
| Py-PTP | 4,4′,4″,4‴-((pyrene-1,3,6,8-tetrayltetrakis(benzene-4,1-diyl))tetrakis(ethyne-2,1-diyl))tetrabenzoate |
| Por-PTP | meso-tetrakis-(4-((phenyl)ethynyl)benzoate)porphyrin |
| EDDB | 4,4′-(ethyne-1,2-diyl)dibenzoate |
| CTTA | 5′-(4-carboxyphenyl)-2′,4′,6′-trimethyl-[1,1′:3′,1″-terphenyl]-4,4″-dicarboxylate |
| TTNA | 6,6′,6″- (2,4,6-trimethylbenzene-1,3,5-triyl)tris(2-naphthoate)) |
| PEDC | 4,4′-(1,4-phenylenebis- (ethyne-2,1-diyl))dibenzoate |
| BTDC | 2,2′-bithiophene-5,5′-dicarboxylate |
| BTBA | 4,4′,4″,4‴-(biphenyl-3,3′,5,5′-tetrayltetrakis(ethyne-2,1-diyl))tetrabenzoate |
| PTBA | 4-[2-[3,6,8-tris[2-(4-carboxylatephenyl)-ethynyl]-pyren-1-yl]ethynyl]-benzoate |
| BTE | 4,4′,4″-(benzene-1,3,5-triyl-tris(ethyne-2,1-diyl))tribenzoate |
| BTPP | 1,3,5-Tris((1H-pyrazol-4-yl)phenyl)benzene |
| BTP | 1,3,5-tris(1H-pyrazol-4-yl)benzene |
| 1,4-BDP | 1,4-benzenedi(4′-pyrazolyl) |
| 1,3-BDP | 1,3-benzenedi(4′-pyrazolyl) |
| TPP | 10,15,20-tetra(1H-pyrazol-4-yl)-porphyrin |
| mIM | 2-methylimidazolate |
| bIM | benzimidazolate |
| nIM | 2-nitroimidazolate |
| 5cbIM | 5-chlorobenzimidazolate |
| ICA | imidazolate-2-carboxyaldehyde |
| 5-mTz | 5-methyltetrazolate |
| 2-mbIM | 2-methylbenzimidazolate |

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
