# Peer review of "Application of Various Metal-Organic Frameworks (MOFs) as Catalysts for Air and Water Pollution Environmental Remediation"

_catalysts, doi:10.3390/catal10020195_

Round 1

Reviewer 1 Report

While the authors have made efforts to remedy some of the previously described problems, the English grammar and style are still vastly below the expected standard for publication. There are sentences throughout the paper which are missing key words, making it impossible to understand what is being described scientifically.

This is even evident in paragraphs that are marked as having been revised, suggesting that no major review of the language in the paper has been conducted by the authors and if indeed any attempt has been made, it is inadequate and requires a thorough rewriting to meet acceptable standards for publication.

On these grounds, I cannot recommend this paper for publication.

Author Response

Thank you for your kind notice and I apologize carelessly the draft. However, we did our best to correct the grammatical errors using the grammar service. Also, we are fully sorry for not meeting your needs.

Reviewer 2 Report

A PDF is attached.

Author Response

Comments to the Authors: The revised version of this manuscript has been improved in numerous aspects in agreement with previous remarks raised by the reviewers (such as title, citations, or graphics). In a broad context, I agree this contribution lies in the scope of Catalysts and the information gathered here on MOFs and environmental remediation is interesting and deserves publication. Yet, the authors should have a careful look at some points. The artwork is certainly a major concern. It’s true that some plots have been improved and the resulting figures show homogeneous formats;

Comment 1: Unfortunately others do not. The problem comes from original figures reproduced directly without editing (e.g. Figures 5, 6, 7, 8, 9, 14, and Scheme 3), having significant disproportion between text and the rest of the image(s), or different parts within the same image (e.g. Scheme 3). Thus, legends on the X and Y axis, labels (a), (b), etc. are shown with different sizes and fonts. I would recommend edition in either ppt(x) or PDF formats and then saved as new images incompatible format. Like a review, I realize the authors have taken all graphics from the original sources.

Response: Thank you for your kind suggestion. We re-drew the figures 5, 6, 7, 8, 9, and 14. However, we don’t need to re-drafted Scheme 3 due to make that ourselves. (Please see: Manuscript Page 11, 12, 13, 14, 15, and 20.)

Comment 2: As already mentioned, this does require permissions explicitly granted by the publisher, unless the paper in question is published under a Creative Commons Licence with specific conditions. While permissions have apparently been granted for some figures, others lack this information (Figures 4-6 and Schemes 1-3).

Response: We fully agree to the reviewer’s comment. We get copyright from the original publisher and re-drafted the caption of figure 4-6. However, we don’t need to permission about schematic images due to make that ourselves. (Please see: Manuscript Page 10 Figure 4, Page 11 Figure 5, and Page 12 Figure 6, respectively.)

Comment 3: The legend for permissions should be typed according to the specifications of Catalysts, but in general, the format is: Reproduced and/or adapter from Ref. [xx]; Copyright (year) + Publisher. As noted in my former comments too, Summaries after every subsection is rather repetitive; some statements have been said in previous paragraphs, while other ideas seem to be new. Summaries are fine for original research, not for reviews.

Response: Thank you for your kind suggestion. We have re-drafted the all figure’s caption. (Please see: Manuscript 6, 8, 11, 12, 13, 14, 15, 18, 19, 20, and 21 Page 16, Figure 1-14.)

Reviewer 3 Report

This is a very interesting and important review describing the role of various metal-organic frameworks  as catalysts for air and water pollution
MOFs have many advantages, such as high specific surface area, rich topology, and easily tunable porous structure.  
Environmental remediation by MOFs gave rise to great interest in this area and stimulated the investigations on wastewater treatment, air purification, and disinfection.
 However, together with the practical successes and active discussions about the mechanism and synthetic method of composites, the scientists began to realize that they tackle the complex systems which comprise the particles of different nature.
Of course, it is impossible to cover all the works in one review, the authors made their own choice which sometimes does not coincide with the viewpoint of the reviewer, but I think that it is their right.
My small remarks:
1)  It is also worth presenting other ways to coordinate pyrazole ligands ( see  J. Am. Chem. Soc. 2012, 134, 30, 12830).
2) Is there no other Pd@MOF material used to reduce of nitrophenol? 

Author Response

Comment 1: It is also worth presenting other ways to coordinate pyrazole ligands (see J. Am. Chem. Soc. 2012, 134, 30, 12830).

Response: Thank you for your kind recommendation and suggestions. We have added in reference to what you recommend. (Please see: Manuscript Page 1, Line 38 and Manuscript Page 2 Line 52, 65)

Comment 2: Is there no other Pd@MOF material used to reduce of nitrophenol?

Response: Thank you for your kind suggestions. We searched about Pd@MOF material used to reduce of 4-nitrophenol provided. Unfortunately, they didn’t exhibit the recyclable tests and not good characterization about catalysts. (Please see: Wang, C.; Zhang, H.; Feng, C.; Gao, S.; Shang, N.; Wang, Z. Multifunctional Pd@MOF core-shell nanocomposites as a highly active catalyst for p-nitrophenol reduction. Catalysis Communications 2015, 72, 29-32)

Round 2

Reviewer 2 Report

The second version of this manuscript has been improved further. I would be in favor of publication, given the potential usefulness of this information on MOFs and their applications. Please consider the following remarks:

Introduction (pp. 1-2): why references on MOFs jump from 5-8 to 220 [5-8, 220]? I realize the insertion of a new citation involves re-numbering; please delete or modify. Legends for copyrighted work: Contact the editorial office for a preferred style, if any. The copyright is usually owned by the publisher, not the author (delete this at the end). Also: “Reproduced and adapted” from (not adapter, sorry for this misspelling in my former report). In my opinion, as emphasized previously, summaries are unnecessary in this review article.

Author Response

Comment 1: Introduction (pp. 1-2): why references on MOFs jump from 5-8 to 220 [5-8, 220]? I realize the insertion of a new citation involves re-numbering; please delete or modify.

Response: Thank you for your kind notice. We have re-drafted the reference number. (Please see: Manuscript Page 1 and 2 Line 38, 52, and 65)

Comment 2: Legends for copyrighted work: Contact the editorial office for a preferred style, if any. The copyright is usually owned by the publisher, not the author (delete this at the end). Also: “Reproduced and adapted” from (not adapter, sorry for this misspelling in my former report).

Response: Thank you for your kind notice. We have re-drafted the each caption for copyright. (Please see: Manuscript Page 7, 9, 11, 12, 13, 14, 15, 17, 18, 19 and 20 Figure 1, 2, 3, 4, 5, 6, 7, 8, 9, 10, 11, 12, 13, 14 and Scheme 4, respectively.)

Comment 3: In my opinion, as emphasized previously, summaries are unnecessary in this review article.

Response: Thank you for your kind opinion. We removed summary of each part.

This manuscript is a resubmission of an earlier submission. The following is a list of the peer review reports and author responses from that submission.

Round 1

Reviewer 1 Report

In this review paper, the authors try to illustrate some important works that focus on MOFs for environmental remediation. A variety of topics from gas sensing to water pollutent detection, and their capture were included. The authors also introduced fabrication of MOF-polymer fiber composite and their application in air pollution control.

Overall, I can see that the authors did a massive amount job to summarize the literature, which is of great help to gain basic knowledge of MOFs. However, as a review paper, it is somehow lack of arrangement as well as correlation between different sections, also the same case for the parts within the section. It is too preliminary to be published. So I recommend the authors to revise and rearrange the current manuscript, narrow down the topic, and give more specific reviews and discussions. I would love to review it again if the authors have chance to resubmit their revised version.

Also some suggestion to the authors:

MOFs for environmental remediation is too big a topic that it can become a book other than a review paper. I would recommend the authors to choose one of the directions and go deep. The abstract part has several grammar errors. I suggest to rewrite this part. Also the keywords cannot reflect the main topic of the review paper, as a paper discussing environmental applications, keywords like hydrogen storage is not appropriate. In the introduction part: three paragraphs introduce nitroarene, SO2 sensing, and ORR in detail, but not concentrate to the main topic. When using the figures from literatures please get the copyright from the original publishers.

Reviewer 2 Report

This has severe issues with English language and grammar throughout, with 3 errors in the abstract alone. The errors persist throughout the paper. Extensive proofreading and editing would be required to make the paper suitable for publication. If the journal offers a service to help where someone with suitable proficiency is unavailable, I recommend using it.

Further to this, the use of citations in the paper is inconsistent and the citations are often placed inappropriately or missed out entirely at the relevant points throughout the article. There are entire paragraphs in Section 3 that talk about a study without actually giving the reference they talk about. The figures that have been used are not properly cited and this is prevalent throughout the paper. There are also references to values that are not compared with other materials or given any context other than they are "good".

Several of the figures are of extremely low quality with Figure 5 being the worst offender. Some also lack suitable legends and in some cases the figure captions, particularly in Section 5, do not describe what the figure is showing. This section is also generally irrelevant to the papers purpose to show catalytic change of environmental contaminants, of which hydrogen is not one.

The electrospinning procedure is interesting but lacks details. Unless these are being witheld for intellectual property reasons, I suggest giving more detailed descriptions of the process, conditions and the reasoning behind how controlling voltage etc. can change the end-product. Section 6, if properly expanded could make an interesting paper in its own right and should be considered if not for this journal, then a suitable journal which discusses methods.

Due to the above issues, I cannot recommend this paper for publication and suggest it be entirely reworked.

Reviewer 3 Report

This comprehensive article reviewes the use of some MOFs, often in the context of nanotechnology, to solve enviromental concerns such as detoxification, filtration, or sensing of polluting gases. The catalytic activity of such MOFs and their derivatives is nicely illustrated. Although I support publication of this valuable contribution, some revisions (both major and minor) should be addressed prior to acceptance.

Probably, the first concern (an important issue) is that most (if not all) figures are taken from original sources without crediting them. Note that, due to ethical considerations, such reproductions require the corresponding permissions. I asume the authors are familiarized with this point as copyrights to reproduce a figure/scheme from a journal to reuse in another journal can easily be obtained, e.g. via RightsLink at the journal’s website. All legends must incorporate a final sentence acknowledging explicitly the source and/or statement required by the publisher.

Also, the fact that figures are directly copied and pasted with different sizes, fonts, and scales leads inevitably to a non-aesthetic artwork. I would suggest some editing in the search of homogeneity. Note for instance, acronyms in Tables and figures, legends at X- and Y-axis, etc.

Every section/subsection concludes with a summary (albeit the length ranges from one sentence to long paragraphs). In principle, this strategy appears to be useful. However, the summaries often provide additional info and data not collected in the preceding text, which may mislead the readers. Moreover, this makes the manuscript unnecessarily long. I would rather remove those concluding statements, incorporating specific data, if pertinent, into the above paragraphs.

Although for the purpose of this review the background on MOFs (Sect. 2: Basics) suffices; further structural data could be of interest to the readership, such as dimensions of unit cells or inner cavity sizes, to understand better host-guest interactions (e.g. Figures 1-3).

Taking into account that this paper focuses on environmental impact, this reviewer wonders whether toxicity data of such MOFs are available. The integration of polluting ions, like Cd(II), into the framework raises some concerns, even if used in catalytic cycles.

Schemes vs Figures (unclear too). If the intention is denoting a chemical reactions as scheme, then this should strictly be applied to Schemes 2 and 3 only.